# Challenges in Assessing Aphasia in Congenital Blind Patients: A Case Report

**DOI:** 10.3390/reports6040049

**Published:** 2023-10-11

**Authors:** Maria Grazia Nicoletta, Francesco Riganello, Lucia Francesca Lucca, Maria Daniela Cortese

**Affiliations:** Istituto S. Anna, Research in Advanced Neurorehabilitation University, 88900 Crotone, Italy; mg.nicoletta@isakr.it (M.G.N.); francescoriganello@gmail.com (F.R.); l.lucca@isakr.it (L.F.L.)

**Keywords:** broca aphasia, sensory aids, communication aids for disabled, blindness, health communication

## Abstract

This case report focuses on assessing aphasia in a congenitally blind patient with an ischemic lesion using the Aachener Aphasia Test. The method involved adapting existing assessment tools to the patient, integrating Braille as an accessible technology, and incorporating the patient’s family for emotional support and for the identification of patient-specific communication strategies. The assessment revealed patient strengths in areas such as articulation, prosody, and repetition skills, but also exposed challenges in semantic and syntactic structures. However, the unavailability to assess and score naming and comprehension limited a full assessment of the patient’s language abilities. The findings underscore the need for flexible, tailored assessment strategies and collaborative approaches involving healthcare professionals and families. Moreover, it suggests a considerable research gap and a need for standard tools to assess blind patients with aphasia comprehensively. This case report contributes to the limited knowledge of assessing aphasia in blind individuals and calls for further research in this area to refine and expand the available tools and strategies.

## 1. Introduction

Aphasia is a language disorder that impairs speech, understanding, reading, and writing as a result of damage to the brain areas responsible for language processing, usually located in the left hemisphere [1,2]. This condition can affect all aspects of language processing and may be accompanied by other neurological symptoms such as dysarthria, apraxia, hemiparesis, hemianopsia, or dysphagia, particularly in stroke patients [3,4].

Stroke is the primary cause of aphasia. Other causes include head injuries; brain tumors; and various inflammatory, infectious, toxic, metabolic, and degenerative diseases.

The incidence of stroke rises with age and affects men more frequently than women [5,6,7]. Approximately 40% of all stroke survivors develop aphasia, especially those who have had a cardioembolic stroke or undergone thrombolysis.

Post-stroke aphasia (PSA) is a consequence of an injury to an extended network of cortical and subcortical structures perfused by the middle cerebral artery in the left hemisphere [8]. This condition can lead to long-term morbidity, mortality, and residual disability [9], impacting an individual’s ability to communicate and participate in social activities [10].

PSA results have an incidence of around 151 per 100,000 people per year [11], with an incidence within ischemic patients constant at 30% [12]. It is more common among older individuals (i.e., 15% of individuals under 65 y.o. experience aphasia after their first ischemic stroke), with a percentage that increases to 43% for individuals older than 85 years [13].

Similar to other high-income countries, stroke rates in Italy range from 1.8 to 4.5 new cases per 1000 people per year, with a prevalence of 6.5 percent [5].

The severity of aphasia can vary, from mild cases with occasional word-finding difficulties to complete loss of verbal communication, and can also fluctuate over time, with some aspects of language impairment improving while others remain affected.

Aphasia is commonly categorized into two primary types: fluent and non-fluent [14]. Both fluent and non-fluent types of aphasia are reported in acute cases (less than 4–6 weeks post-stroke), with a higher number of fluent aphasias in chronic cases (i.e., one year or more post-stroke).

Fluent aphasia, similar to Wernicke’s aphasia, is typically characterized by difficulties in comprehending spoken language, while the ability to produce connected speech remains relatively intact. However, fluent speech can still be abnormal, with disjointed sentences and the intrusion of unrelated words, sometimes resembling jargon in severe cases. Reading and writing abilities are often significantly impaired in this type of aphasia.

On the other hand, non-fluent aphasia, such as Broca’s aphasia, is characterized by a significant reduction in speech output, with speech often limited to short phrases of fewer than four words. Individuals with Broca’s aphasia may struggle with vocabulary access and exhibit laborious, clumsy sound formation. While they may understand spoken language reasonably well and be able to read, their writing abilities are often limited. Although anomia is a core symptom in all aphasic syndromes, agrammatism and apraxia of speech serve as clinical markers to distinguish Broca’s aphasia from other forms of the disorder.

As linguistic and non-linguistic processes, such as attention, memory, and sensory or motor functions, are interconnected and contribute to language abilities, aphasia is a multifaceted disorder requiring an appropriate assessment method. This comprehensive approach is crucial for developing effective rehabilitation strategies and minimizing the impact of aphasia on daily life and professional activities.

Given the high incidence of linguistic deficit in post-stroke patients, some may have a premorbid or acquired sensory deficit such as blindness.

The worldwide blind population is estimated to be around 43 million (0.5%), and an additional 295 million (3.7%) suffer from moderate to severe vision impairment [15]. Even though the chances are low for a subject experiencing blindness to also suffer from a stroke and aphasia, when such a case does occur, their evaluation and rehabilitation become notably challenging.

The Aachen Aphasia Test (AAT) is a standardized tool widely used for diagnosing and evaluating different types of aphasia, such as Broca’s aphasia [16]. AAT is a well-established neuropsychological tool designed to diagnose and quantify aphasic deficits, primarily in individuals who have suffered cerebral injuries such as strokes. AAT consists of multiple subtests that focus on different aspects of language: spontaneous speech (i.e., communication, articulation, prosody, automatic speech, and semantic and syntactic structures), auditory comprehension, repetition, naming, reading, and writing. Each subtest is scored separately, and the combined scores provide a comprehensive overview of an individual’s language skills. This aids in identifying the type of aphasia and assessing its severity. AAT subtest scoring ranges differ among the subtests, with the *Token test* and *written language* scored ranging between 0 to 50 and 0 to 90, respectively; the *repetition* item score ranges from 0 to 150, while other items range from 0 to 120. Trained clinicians typically administer AAT, and it takes between 60 and 90 min to complete. In addition to its diagnostic capabilities, AAT is also used for therapeutic planning and for monitoring the effectiveness of rehabilitation.

However, because AAT depends on visual stimuli such as the token test, naming, and comprehension, it is not suitable for blind patients.

Assessing aphasia in blind patients presents unique challenges. While AAT is widely recognized and utilized, its reliance on visual stimuli renders it less suitable for blind patients. Consequently, there is a risk that assessment using AAT may yield incomplete or distorted information when applied to this demographic. Moreover, while involving the family can be essential for accurately assessing a blind patient’s linguistic abilities, there remains a significant gap in the literature. Currently, no work has been found on adapting aphasia assessment tests tailored specifically for the blind population.

A work of Smith on visual field defect and aphasia testing proposed an adaptation of the Boston diagnostic aphasia examination test [17], but no adaptation for the aphasic test in the case of blind subjects was found.

Other works studied the different issues when assessing blind patients with aphasia. Birchmeier [18] reported a case of severe expressive aphasia (Wernicke’s aphasia) in a congenitally blind 82-year-old man following a stroke. It affected his left hemisphere with impairments in speech production, naming, repetition, and writing, as well as alexia and dyslexia when reading Braille. This case highlights the challenges of assessing language abilities in blind individuals, the specificity of braille alexia, and the interactions between perceptual and linguistic processes in reading. In another case report, Parker [19] described the treatment of a blind, 73-year-old woman with Wernicke’s aphasia following a stroke. In addition, in this case, the therapist had to creatively adapt standard visual techniques such as matching exercises and written cues for the patient. Therapy focused on braille reading and writing to improve the patient’s expressive skills. The used approach and the patient’s motivation support allowed for good patient’s recovery. In this work, Parker highlighted the inherently visual typical aphasia therapy and the need to develop non-visual adaptations to serve patients with sensory deficits.

The primary aim of our work is to describe the strategy we adopted to assess aphasia in a blind patient, offering possible insights and reflections on the work with this unique patient group.

## 2. Detailed Case Description

The patient is a 69-year-old retired individual with congenital blindness, married with sons, with a medium/high level of education (university degree). Before the stroke, he was fluent in both spoken language and Braille, showcasing advanced linguistic skills consistent with his medium/high level of education (holding a university degree). He was self-sufficient, commuting from home to work on foot with the aid of a walking stick. Additionally, he was proficient in using voice-command apps, for assistance, further demonstrating his linguistic adaptability.

After being admitted to the emergency department of the nearest hospital facility, upon CT the following was diagnosed: occlusion of the apex of the left carotid siphon, with marked opacification of the segment of the Middle Cerebral Artery and of the proximal portion of the Anterior Cerebral Artery (Figure 1).

The patient was hospitalized 15 days after the acute event. At the first assessment, the patient was alert and cooperative, with symptoms consistent with a diagnosis of predominantly Broca’s aphasia, characterized by fragmented speech with a relatively preserved intelligibility, possibility of understanding short phrases, responses that were not always consistent with the content of the question, and echolalia. The patient also demonstrated relatively preserved repetition of words and phrases. The patient’s blindness did not allow for standard AAT administration.

### 2.1. Speech Therapist and Family’s Role in Patient Assessment

To rigorously tailor the assessment and rehabilitation to the patient’s condition, his family’s involvement in the therapeutic process was strategic. Family members actively participated in the assessment process, assisting in tasks requiring personal knowledge about the patient’s pre-stroke linguistic abilities and daily communication habits. Their insights were invaluable, especially when adapting assessment tools for the patient.

The approach to the patient had two complementary aspects: the roles of the speech therapist and the family (Figure 2).

Speech therapist’s role:Adapt existing assessment tools: modify the AAT by using tools designed explicitly for blind patients.Integrate technology: utilize accessible technology, such as Braille displays, to assist in assessment and intervention.

Family’s role:Provide background information about the patient’s history and preferences.Identify patient-specific communication strategies: provide information on the patient’s pre-stroke communication abilities and compensatory strategies, such as reliance on auditory cues and tactile feedback.Offer emotional support: become involved and encourage the patient during the assessment process to help alleviate anxiety and create a supportive environment.

### 2.2. AAT Modification and Patient’s Assessment

The Italian version of AAT, adapted from the original German AAT [20], remains a primary diagnostic tool for evaluating acquired linguistic impairments. In Italy, it is fundamental in clinical settings for assessing aphasia. The Italian version of AAT provides a probabilistic analysis of aphasic challenges, employs inferential statistical methods to gauge individual performance, and offers benchmark data for an in-depth analysis of aphasia cases in research [16].

AAT’s strength lies in its comprehensive approach. It evaluates multiple linguistic facets, both in oral and written forms, encompassing auditory and visual comprehension and expression. The subtests consist of three to five sets, each with ten items, targeting varied linguistic elements—from phonemes to complex lexemes and sentences. Each item is systematically arranged based on linguistic intricacy, ensuring the test can discern aphasic conditions across varying severity levels.

Besides the oral items that assess a person’s proficiency in understanding and articulating spoken language, AAT typically integrates tasks involving visual cues. Patients are shown images or words, requiring them to complete several activities such as object naming, word−picture matching, or adhering to written instructions. These visual cues are pivotal for assessing both receptive and expressive language capabilities.

For our blind patient, we used specific adaptations (Table 1). The written language assessment was tailored using Braille (Figure 3). In this phase, interaction with the patient’s family was essential, which transcribed parts of the test regarding the reading ability in Braille. The written language assessment utilized a Braille display comprising raised pins or dots felt by touch. Braille letters consist of combinations of up to six raised dots. Blind individuals can input using various methods, including a standard or specialized Braille keyboard. In this instance, a keyboard with Braille-dot-corresponding keys facilitated the patient’s input.

### 2.3. Outcome

Considering the uncompromised tactile sensitivity and the patient’s propensity toward smelling manipulable objects, we adapted the “picture naming of objects with simple names” and “picture naming of objects with compound name” items for the naming AAT subtest. Typically, it consists of three groups of ten objects, with ten focusing on color recognition. For our patient, we used 20 tangible objects.

The comprehension was tested by asking for the execution of simple orders, but no score was assigned.

As in AAT, spontaneous language was assessed by communicative ability; articulation and prosody; automatic speech; and semantic, phonemic, and syntactic structures.

The adapted assessment approach provided valuable insights into the patient’s language abilities, revealing strengths and weaknesses in various language domains (Table 2).

The assessment of spontaneous language facilitated the evaluation of the patient’s communicative ability; articulation; prosody; automatic speech; and semantic, phonemic, and syntactic structures.

The comprehension assessment, involving repeating simple orders, revealed the patient’s ability to understand and follow instructions to a certain extent.

Using a Braille display enabled assessing competence in written language.

Finally, adapting the naming test, which involved using natural manipulable objects, was effective in assessing the patient’s naming abilities.

During the AAT assessment, in the spontaneous language section, the patient exhibited difficulties with semantic and syntactic structures, while performing relatively well in automatic speech and phonemic structure. The best results were observed in articulation and prosody.

He demonstrated a good ability to repeat sounds, words, and sentences in the repetition test. The performance was good for written language, specifically in reading and dictation by composition when using Braille.

Given the blind patient’s condition, adapting the naming test for this patient using manipulable objects was effective, but no score was assigned to colors and figure description items. Similarly, no score was assigned to comprehension. However, the patient’s ability to understand and follow simple orders was observed.

The results highlighted the patient’s strengths in certain areas, such as articulation, prosody, and repetition skills. However, they also underscored the challenges in semantic and syntactic structures. The use of Braille was proven to be effective in specific areas of written language but was limited in others, such as dictation in handwriting.

The impossibility of assigning a score in comprehension and specific areas of naming items made it difficult to assess the patient’s language abilities fully.

Overall, the assessment indicated a slight deficit in repetition and a moderate deficit in the written language and naming (denomination of the objects) items (Figure 4). However, the assessment result did not fully describe the patients because of the impossibility of assessing the token test, comprehension, and part of the naming items.

## 3. Discussion

The results emphasize the need for a tailored approach in assessing language abilities of congenitally blind patients. The standard use of AAT might not be fully applicable or not capture the full spectrum of the patient’s abilities and challenges.

Furthermore, this case report highlights the crucial role of collaboration among healthcare professionals, family members, and the patient when assessing aphasia in congenitally blind patients.

The tailored approach to the patient’s assessment, which included adapting existing assessment tools, collaborating with a team of experts, integrating technology, and involving the patient’s family, proved effective when evaluating the patient’s language abilities and informing the subsequent rehabilitation process.

This work suggests a potentially valuable approach for blind patients with aphasia, emphasizing the need for flexible and adaptable assessment strategies. Moreover, this case report contributes to the limited body of knowledge on assessing aphasia in blind individuals, underscoring the importance of considering individual patient characteristics and needs when developing assessment and intervention plans.

However, the lack of a standardized tool tailored to this patient population makes assigning accurate scores and conducting a thorough assessment challenging.

Research has shown differences between sighted and blind individuals. Studies in sighted humans report a consistent left-lateralized network of brain regions involved in language processing, including areas of the frontal, temporal, and parietal lobes [21,22]. Nevertheless, in blind individuals, there is an additional recruitment of visual cortical areas for language tasks [23,24,25] (i.e., congenitally blind subjects show visual cortex activation including V1, during Braille reading, verb generation, word processing, and sentence comprehension).

Nevertheless, no studies were found regarding the issues faced when assessing aphasic blind patients.

Assessing aphasia in blind patients necessitates a broader sensory perspective, particularly emphasizing tactile [26,27] and olfactive [28] channels.

For example, Braille letters consist of combinations of up to six raised dots, and the brailler has six keys, three for each hand, which are operated in various combinations. The difficulty is not only the coordination of fingers of the right hand, but also considering possible typical dysgraphic reversals and omissions of letters. That is, where the sighted aphasic may confuse “b”, “d”, “p”, and “q”, the blind aphasic may confuse: “d”, “f”, “h”, and “j”.

The importance of the olfactive sensorial channel was observed in naming the object. We presented the necessity of working with real objects (e.g., real fruit and not plastic representation) to touch and smell them.

The integration of these sensorial channels will not only enhance the accuracy of assessments, but also provide a richer, more holistic understanding of the patient’s linguistic abilities and challenges.

Considering the high incidence of language disorders following acquired brain injury, the described cases underscore the lack of tools and helpful information for specific deficits associated with sensory impairments such as blindness. The speech therapist’s ability to adapt an assessment to the patient’s needs while adhering to rigorous theoretical models is crucial.

Birchmeier [18] and Parker [19], in their works, emphasized the complexities of evaluating language skills in blind people, underscoring the visual nature of typical aphasia therapy and the imperative for non-visual adaptations for those with sensory impairments.

This report outlines our approach to assessing a blind patient diagnosed with Broca’s aphasia. While we are not suggesting changes to AAT, we aim to highlight the noticeable void in the current literature regarding this distinct patient demographic. Our study emphasizes the need for future works to enhance both assessment and therapeutic strategies for aphasic blind individuals. Collaborative endeavors with families and institutes for the blind are essential for developing standardized tests specifically designed for this patient group.

## Figures and Tables

**Figure 1 reports-06-00049-f001:**
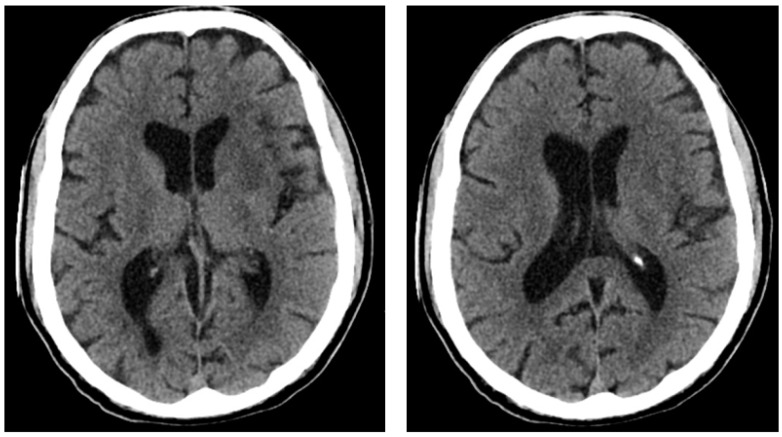
Computerized axial tomography of the patient. Asymmetric ventricular system in the supratentorial area, predominantly on the right and on the midline. General volume increase in the subarachnoid spaces at the top and base. Subtle hypo-density of the left insulo-frontal cortico-subcortical region, partially extending to the corresponding nuclear-capsular region, as if from a recent ischemic lesion.

**Figure 2 reports-06-00049-f002:**
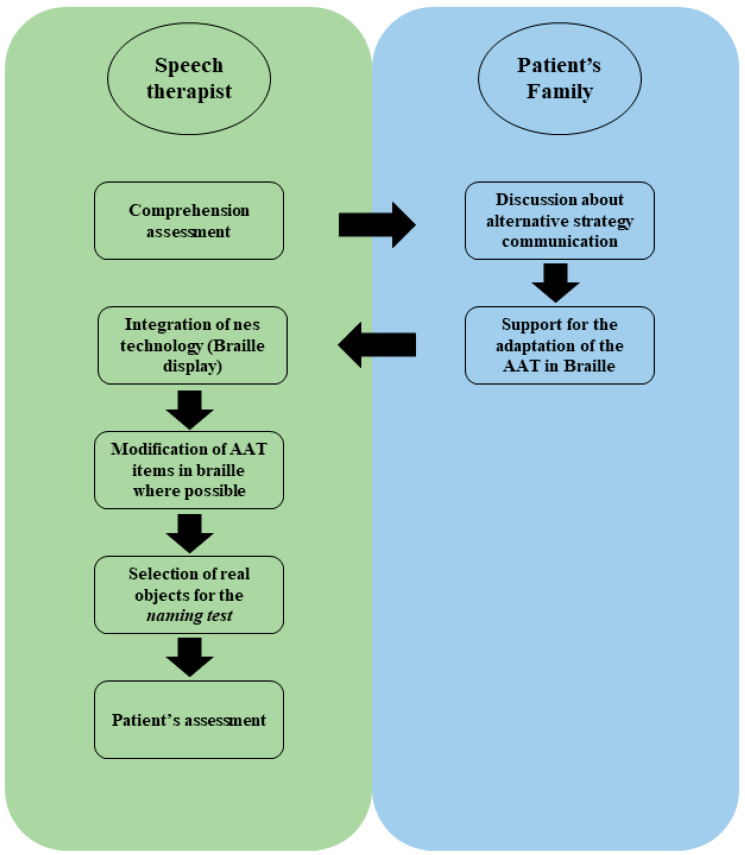
Scheme of the adaptation process of the AAT test to blind patients. After the first approach to the patients, the speech therapist interacted with the patient’s family to study his communication modality and define the strategy for a possible correct approach to the assessment and AAT administration. The patient’s family collaborated with the speech therapist to translate part of the AAT test regarding written language in Braille. For the naming test, we studied which natural object to use for the item regarding the denomination of the objects. After defining the modification to the patient’s AAT test, the test was administered for the patient’s assessment.

**Figure 3 reports-06-00049-f003:**
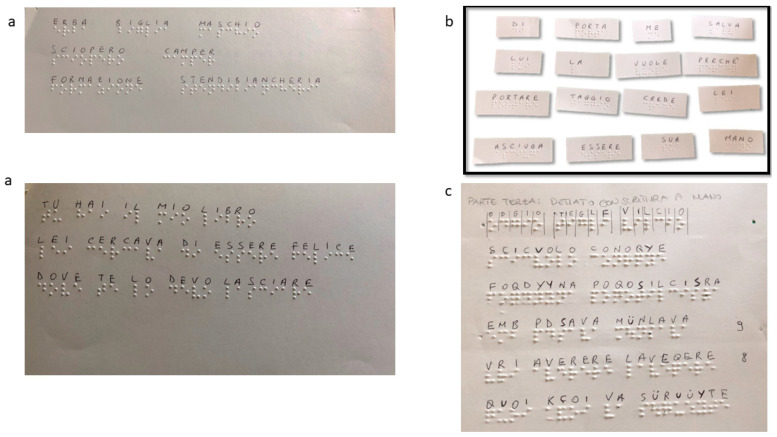
Example of adaptation of AAT. (**a**,**b**) Reading and Composition—the family’s patient provided the words and phrases in AAT in Braille, allowing for the administration of part of the written language items. (**c**) Dictation—The patient used the Braille display tool to write in Braille.

**Figure 4 reports-06-00049-f004:**
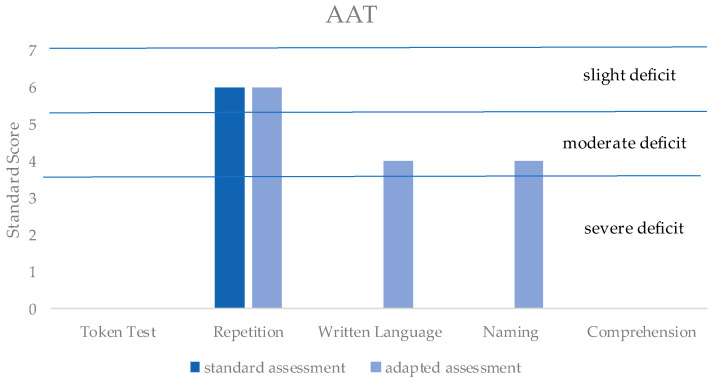
Based on the patient’s score in the different AAT items, the standard score defines the severity level of the deficit for each item (i.e., 1–3: severe deficit; 4–5: moderate deficit; 6–7: slight deficit; 8–10: no deficit). In dark blue is the score that the standard AAT administration would have obtained without the adaptation of AAT; in light blue is the score obtained with the AAT-adapted administration. It is important to note that no adaptation was possible for the token test and comprehension, while the naming item was partially adapted.

**Table 1 reports-06-00049-t001:** Scheme of the AAT test and relative adaptation to the patient.

	Observation Levels	Description	Adaptation	Total Range by Subtest
Token Test *		Assesses multiple aspects of language comprehension based on the ability to identify colors, shapes, and verbal commands. In the Token Test, the number of errors is counted.	n/a	0–50
Repetition	- single phonemes	Assesses the ability to repeat sounds, words, less common words, phrases, and sentences.	Standard administration	0–150
	- one–three syllabic words			
	- one–three syllabic loanwords			
	- noun phrases of increasing length			
	- sentences of increasing length			
Written Language	- reading aloud words and sentences (Figure 3a)	Measures reading ability	The Braille display was provided to the patient to write in Braille during the test	0–90
	- composing words/sentences from graphemes/morphemes (Figure 3b)	Assesses the ability to write from dictation		
	- writing words/sentences to dictation (Figure 3c)	Measures handwriting skills		
Naming	- picture naming of objects with simple name **	Assesses the ability to name objects	Natural and manipulable objects have been used for simple and compound names; the assessment was not possible for “naming colors”	0–120
	- naming of colors	Assesses the ability to name colors		
	- picture naming of objects with compound name **	Measures the ability to name objects with compound words		
	- description of situations and actions	Measures the ability to describe sentences		
Comprehension *	- auditory comprehension of words	Measures the ability to comprehend words and sentences	n/a	0–120
	- auditory comprehension of sentences			
	- reading comprehension of words			
	- reading comprehension of sentences			

* Adaptation not applicable. ** List of objects used for simple (a) and compound (b) names (in parenthesis the correspondent Italian name): (a) apple (mela), pear (pera), banana (banana), orange (arancia), bottle (bottiglia), hammer (martello), pen (penna), brush (pennello), fork (forchetta), glass (bicchiere); (b) nutcracker (schiaccianoci), screwdriver (cacciavite), can opener (apriscatole), wallet (portafoglio), remote control (telecomando), towel (sciugamano), shoehorn (calzascarpe), toothpaste (dentifricio), colander (scolapasta), and squeezer (spremiagrumi).

**Table 2 reports-06-00049-t002:** Aachener Aphasie Test: patient’s assessment.

Spontaneous Language	Method of Administration	Range					
		0	1	2	3	4	5
Communication	Standard		X				
Articulation and Prosody	Standard				X		
Automatic Speech	Standard			X			
Semantic Structure	Standard	X					
Phonematic Structure	Standard			X			
Syntactic Structure	Standard	X					
**TEST**	**Modality of administration**		**RANGE**		**Score**		

**Token Test**	n/a		0–50				
**Repetition**			0–150				
**Total score**					**123**		
Sounds	standard				26		
Words	standard				30		
Loans and foreign words	standard				28		
Compound words and syntagms	standard				22		
Sentences	standard				17		
**Written Language**			0–90				
**Total score**					**22**		
Reading	Braille				22		
Dictation by composition	Braille				0		
Dictation in handwriting	Braille				0		
**Naming**			0–120				
**Total score**					**45**		
Objects	Modified				30		
Colors	n/a						
Compounds Names	Modified				15		
Figures Description	n/a						
**Comprehension**			0–120				
Listening Words	n/a						
Listening Sentences	n/a						
Writing Words	n/a						
Writing Sentences	n/a						

## Data Availability

Data is contained within the article.

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
