# Peer review of "Challenges in Assessing Aphasia in Congenital Blind Patients: A Case Report"

_reports, 2023, doi:10.3390/reports6040049_

Round 1

Reviewer 1 Report

Dear authors and editor,

The manuscript titled "assessing aphasia in a congenitally blind patient with an ischemic lesion using Aachener Aphasia Test " aimed  to  describe the strategy we adopted to assess aphasia in a blind patient, offering possible insights and reflections on the work with this unique patient group.

There are many minor  issues I'd like the authors resolve.

Abstract

1-Change the keywords. Delete the words "AAT", "Congenital Blindness", "Braille", "communication strategies".  Not found in the MeSH (Medical Subject Headings). Change to "sensory aids", "health comunication"

2-The title is appropriate. The authors identify the study design.

Introduction

3-The introduction is appropriate. The authors define the most relevant concepts and contextualise the topic.

Case Presentation Section

4-The most relevant findings of the case are described. Information is provided in an orderly and clear manner. In addition, the difficulties of the problem and the relevance of the study are presented.

Discussion
  • 5-The discussion summarises the results found, as well as interrelating the findings with those of other authors.
References

6-The references are adequate in relation to the journal's standards.

Author Response

Dear authors and editor,

The manuscript titled “assessing aphasia in a congenitally blind patient with an ischemic lesion using Aachener Aphasia Test ” aimed  to  describe the strategy we adopted to assess aphasia in a blind patient, offering possible insights and reflections on the work with this unique patient group.

There are many minor  issues I’d like the authors resolve.

Abstract

1-Change the keywords. Delete the words “AAT”, “Congenital Blindness”, “Braille”, “communication strategies”.  Not found in the MeSH (Medical Subject Headings). Change to “sensory aids”, “health comunication”

  • Thanks to the referee for the comments and suggestions. Now we reported Keywords present in the Medical Subject Headings.
  • Now, in the Keywords, it read: Broca Aphasia, sensory aids, Communication Aids for Disabled, Blindness, Health Communication.

2-The title is appropriate. The authors identify the study design.

Introduction

3-The introduction is appropriate. The authors define the most relevant concepts and contextualise the topic.

Case Presentation Section

4-The most relevant findings of the case are described. Information is provided in an orderly and clear manner. In addition, the difficulties of the problem and the relevance of the study are presented.

Discussion

  • 5-The discussion summarises the results found, as well as interrelating the findings with those of other authors.

References

6-The references are adequate in relation to the journal’s standards.

  • Thanks to the referee for the comments

Reviewer 2 Report

This report presents a methodology to assess aphasia in a congenitally blind patient with an ischemic lesion. Due to Aachener Aphasia Test depends on visual stimuli, the aphasia assessing in blindness presence is a challenger. Thus, this work contributes in the developing of effective aphasia identification strategies.

Following, I list some points to be addressed to improve the manuscript:

1) It is necessary describe in detail the Aachener Aphasia Test. Moreover, the authors must describe the changes on the classical ATT test to understand your contribution.

2) Figure 2 needs more description in the manuscript.

3) Their methodology contribution must be described in detail. For example, some diagrams and figures could help. In addition, I recommend to mention the future work to improve the proposal.

4) A comparison with works using the classic ATT could help to understand the problem.

5) Is it possible to include some quantitative metrics to make the manuscript more technical sound?

6) The manuscript is clear and well written up to the Case Presentation Section. After that, it's complicated to follow and understand the proposed methodology. I suggest a depth revision.

7) Move to the Introduction the related work described in the discussion.

Author Response

Reviewer 2

This report presents a methodology to assess aphasia in a congenitally blind patient with an ischemic lesion. Due to Aachener Aphasia Test depends on visual stimuli, the aphasia assessing in blindness presence is a challenger. Thus, this work contributes in the developing of effective aphasia identification strategies.

 Following, I list some points to be addressed to improve the manuscript:

 1) It is necessary to describe in detail the Aachener Aphasia Test. Moreover, the authors must describe the changes on the classical ATT test to understand your contribution.

  • Thank you to the referee for the suggestions. We now reported in the introduction the AAT description and in the case description how the AAT was adapted for the patient’s assessment. Table 1 (page 6) was added to describe the AAT items and relative adaptation.

          Now, on page 2, it read:

  • The AAT is a well-established neuropsychological tool designed to diagnose and quantify aphasic deficits, primarily in individuals who have suffered cerebral injuries like strokes. The AAT consists of multiple subtests that focus on different aspects of language: spontaneous speech (i.e., communication, articulation, prosody, automatic speech, and semantic and syntactic structures), auditory comprehension, repetition, naming, reading, and writing. Each subtest is scored separately, and the combined scores provide a comprehensive overview of an individual’s language skills. This aids in identifying the type of aphasia and assessing its severity. The AAT subtests scoring ranges differ among the subtests, with the Token test and written language scored from 0 to 50 and 0 to 90, respectively; repetition item scored from 0 to 150, while other items range from 0 to 120. Trained clinicians typically administer the AAT, and takes between 60 and 90 minutes to complete. In addition to its diagnostic capabilities, the AAT is also used for therapeutic planning and monitoring the effectiveness of rehabilitation.

         and on pages 5-6 it read:

  • The Italian version of the AAT, adapted from the original German AAT [17], remains a primary diagnostic tool for evaluating acquired linguistic impairments. In Italy, it is fundamental in clinical settings for assessing aphasia. The Italian version of AAT provides a probabilistic analysis of aphasic challenges, employs inferential statistical methods to gauge individual performance, and offers benchmark data for in-depth analysis of aphasia cases in research [13]. The AAT’s strength lies in its comprehensive approach. It evaluates multiple linguistic facets, both in oral and written forms, encompassing auditory and visual comprehension and expression. The subtests consist of 3–5 sets, each with ten items, targeting varied linguistic elements—from phonemes to complex lexemes and sentences. Each item is systematically arranged based on linguistic intricacy, ensuring the test can discern aphasic conditions across varying severity levels. Besides the oral items that assess a person’s proficiency in understanding and articulating spoken language, the AAT typically integrates tasks involving visual cues. Patients are shown images or words, requiring them to do several activities such as object naming, word-picture matching, or adhering to written instructions. These visual cues are pivotal in assessing both receptive and expressive language capabilities. For our blind patient, we made specific adaptations(table I). The written language assessment was tailored using Braille (figure 3). In this phase, interaction with the patient’s family was essential, which transcribed parts of the test regarding the reading ability in Braille. The written language assessment utilized a Braille display comprising raised pins or dots felt by touch. Braille letters consist of combinations of up to six raised dots. Blind individuals can input using various methods, including a standard or specialized Braille keyboard. In this instance, a keyboard with Braille dot-corresponding keys facilitated the patient’s input. Considering the uncompromised tactile sensitivity and the patient’s propensity toward smelling manipulable objects, we adapted the “picture naming of objects with simple names” and “picture naming of objects with compound name” items for the naming AAT subtest. Typically, it consists of three groups of 10 objects, with ten focusing on color recognition. For our patient, we used 20 tangible objects (table I). The comprehension was tested by asking for the execution of simple orders, but no score was assigned. As in AAT, spontaneous language was assessed by communicative ability, articulation and prosody, automatic speech, and semantic, phonemic, and syntactic structures.

2) Figure 2 needs more description in the manuscript.

  • Thank you to the referee for the suggestion. The figure 2 (now figure 3) was improved for image and description.

 3) Their methodology contribution must be described in detail. For example, some diagrams and figures could help. In addition, I recommend to mention the future work to improve the proposal.

  • Thank you to the referee for the suggestion. We added a figure (figure 2, page 5) to describe the method used to adapt the AAT to the patient, and added a recommendation for future works in the discussion.

        Now on pages 10-11 it read:

  • Our study emphasizes the need for future works to enhance both assessment and therapeutic strategies for aphasic blind individuals. Collaborative endeavors with families and Institutes for the Blind are essential in developing standardized tests specifically designed for this patient group.

4) A comparison with works using the classic ATT could help to understand the problem.

  • Thank you for the reviewer's suggestion to compare our findings with classic AAT studies in these patient populations. However, no such studies were available for comparison. Nonetheless, we have included a comparison between the patient's assessment of the adapted AAT and the classic AAT in Figure 4 (page 9)

5) Is it possible to include some quantitative metrics to make the manuscript more technical sound?

  • Thanks to the referee for the suggestion. Now, in Figure 4, we reported the AAT Standard Score with the relative degree of severity.

6) The manuscript is clear and well written up to the Case Presentation Section. After that, it’s complicated to follow and understand the proposed methodology. I suggest a depth revision.

  • Thank you to the referee for the comment and suggestion.
  • We have now revised the indicated points. From pages 4 to 9, the manuscript was revised to improve the description of the methodology. All the modifications are in red.

7) Move to the introduction the related work described in the discussion.

  • Thank you to the referee for the suggestion. Now, the related works are reported in the introduction (page 3)

Round 2

Reviewer 1 Report

The authors have made an effort to respond to all recommendations. It has taken up all the considerations and I consider that despite the methodological limitations, the authors have been able to take them up so that the readers are aware of them.

Thank you very much for allowing me to review your manuscript.

Reviewer 2 Report

Thank you for addressing all my comments. The report was improved from its previous version. The current version is now clear and more significant.